# Spectrum Based Power Management for Congested IoT Networks

**DOI:** 10.3390/s21082681

**Published:** 2021-04-10

**Authors:** Kedir Mamo Besher, Juan Ivan Nieto-Hipolito, Raymundo Buenrostro-Mariscal, Mohammed Zamshed Ali

**Affiliations:** 1Erik Jonsson School of Engineering & Computer Science, The University of Texas at Dallas, Richardson, TX 75080, USA; 723kedir@gmail.com; 2Department of Telematics, FIAD-Universidad Autonoma de Baja California, Ensenada 22860, Mexico; jnieto@uabc.edu.mx; 3The Faculty of Telematics, Universidad de Colima, Colima 28040, Mexico; raymundo@ucol.mx

**Keywords:** energy consumption, spectrum management, freescale launchpads, congested IoT

## Abstract

With constantly increasing demand in connected society Internet of Things (IoT) network is frequently becoming congested. IoT sensor devices lose more power while transmitting data through congested IoT networks. Currently, in most scenarios, the distributed IoT devices in use have no effective spectrum based power management, and have no guarantee of a long term battery life while transmitting data through congested IoT networks. This puts user information at risk, which could lead to loss of important information in communication. In this paper, we studied the extra power consumed due to retransmission of IoT data packet and bad communication channel management in a congested IoT network. We propose a spectrum based power management solution that scans channel conditions when needed and utilizes the lowest congested channel for IoT packet routing. It also effectively measured power consumed in idle, connected, paging and synchronization status of a standard IoT device in a congested IoT network. In our proposed solution, a Freescale Freedom Development Board (FREDEVPLA) is used for managing channel related parameters. While supervising the congestion level and coordinating channel allocation at the FREDEVPLA level, our system configures MAC and Physical layer of IoT devices such that it provides the outstanding power utilization based on the operating network in connected mode compared to the basic IoT standard. A model has been set up and tested using freescale launchpads. Test data show that battery life of IoT devices using proposed spectrum based power management increases by at least 30% more than non-spectrum based power management methods embedded within IoT devices itself. Finally, we compared our results with the basic IoT standard, IEEE802.15.4. Furthermore, the proposed system saves lot of memory for IoT devices, improves overall IoT network performance, and above all, decrease the risk of losing data packets in communication. The detail analysis in this paper also opens up multiple avenues for further research in future use of channel scanning by FREDEVPLA board.

## 1. Introduction

The exponential growth in the interconnection of countless Internet of Things (IoT) applications has led to high density communications in IoT network [1], leading to network congestion problems. Furthermore, its high dependence on license-free bands have made spectrum resources increasingly scarce, and resulted in extremely busy channels [2]. This causes the battery power of IoT devices to degrade lot faster than expected; which is directly responsible for the longevity of the IoT network. In addition, due to lack of appropriate spectrum management, while sitting in a routing queue due to congestion delay, there is an increasing risk that the channel power may run out and important IoT data may get lost in transmission. Thus, it becomes of utmost important to pay attention to these problems; energy consumption and spectrum management.

In IoT, Figure 1, generally a sensor is placed near or around the object to be monitored remotely to collect data. The collected data by sensor is included to form an Internet Protocol (IP) packet and sent to the Coordinator Node (CN) by using a transmitter in the sensor device [3,4,5]. This packet sending process should be carried out in a very short time. However, the huge amount of packet flow present in an IoT network environment is the main issue for longer sending time. This leads to a high consumption of power with busy channel problems in IoT networks. Currently, these problems have no effective solutions. The sensor device or data collector itself does not understand how to do spectrum management while in communication. Additionally, with a conventional network setup, the IoT network resources like battery life and communication channels usually do not get any special attention to improve overall power consumption and quality of network. Since IoT networks frequently become congested, the spectrum management and faster degradation of battery life issues for IoT devices must be addressed in order to increase the performance and network lifetime of these small devices.

Different authors proposed some solutions to address these problems. Authors of the reference [6] proposed an intelligent opportunistic routing protocol (IOP) using a machine learning technique to select a relay node from the list of potential nodes to achieve energy efficiency and reliability in the network during packet transmission. The authors of [7] proposed a solution for large-scale heterogeneous WSN networks which consists of a robust clustering mechanism for energy optimization. In this mechanism, nodes declare themselves as a cluster head (CH) based on available resources such as residual power, available storage and computational capacity. The authors of [8] propose a software-defined wireless sensor networking architecture to improve energy efficiency. The purpose of the solution is to reduce the number of packets transmitted by the sensor nodes of the network with less energy resources available and thus increase the lifetime of the sensors. All of the above mentioned works use IoT devices to process there proposed method which by itself consumes a lot of energy, memory and processing capacity of IoT devices. However, our implementation runs entirely from the Freescale Freedom Development Board (FREDEVPLA).

Therefore, in this paper we propose a different solution to manage communication channels in IoT networks, which outperforms the traditional way. We propose a spectrum based power management solution that scans channel, select the least congested channel and uses it for sending packets. It effectively manages power utilization by minimizing power used for retransmission at a connected state of the channel.

Figure 1 shows a typical IoT scenario in a congested network. Data collected by different IoT sensors at different sources use network routing systems to be released to their end destination. The data packet’s source could be from smart city, smart agriculture, social network data, healthcare data, etc. As the data packets reach the channel coordinator (green box in Figure 1), the control data are forwarded to their destination. The coordinator node cannot manage resources because it does not have channel condition information and follows the default way of resource management which does not include any power or spectrum management mechanism.

Using Figure 2, our proposed spectrum based power management architecture for an IoT is explained. The FREDEVPLA is used only to manage spectrum related information between each CN and its IoT devices, such as: when to do the channel scan, which coordinator can do the scan, select a channel with low interference, identify level of congestion based on received LQI, store child table, and decide when to switch a channel for a specific CN.

In our proposed solution, we scan channel conditions when queue delay time is above the limited estimated threshold time and allocate the least congested channel for data communication between IoT device and its CN. The major motive behind this work is that extending IoT device battery life by reducing retransmission created due to busy channel and manage IoT device configurations from a central FREDVPLA device. It effectively manages channel and reduced retransmission which lead to less power consumption. We use a FREDEVPLA device for managing channel status. While managing congestion and coordinating channel allocation at the FREDEVPLA level, our system configures MAC and physical layer of IoT devices such that it provides the outstanding power utilization compared to the basic IoT standard in active state.

Next, we will present some related works in Section 2. Detail functionality of our proposal will be explained in Section 3. In Section 4 and Section 5 we will go through our test set up and the test result analysis, respectively. Result comparison is presented in Section 6, and finally in Section 7, we conclude our research.

## 2. Related Work

As the standard for IoT is currently in the development phase, power-efficient communication techniques are in demand to facilitate long-lasting battery life on IoT devices. At least ten years battery life is expected in order to minimize the overall maintenance cost [9,10]. So, to extend the battery life of IoT devices it is important to decrease the packets delivery time delay. Lower latency improves power wastage caused by retransmission and wait time [4,11]. Furthermore, solving problems like retransmission and delay without having good spectrum management in congested IoT environment is almost impossible. Similarly, a billion varieties of IoT packets are being sent to the routing system, the default coordinator process method cannot identify the best channel while communicating [12,13]. Wireless sensors in IoT have limited resources in terms of memory and data processing capability. That is why most of the methods proposed to implement spectrum management at IoT device level are not successful. It is proved by multiple studies [3,9,14,15,16,17,18,19] that use of multichannel not only improve the performance of the network but also lower power consumption of the overall network by significant amount.

Currently, there are limited power management methods from spectrum perspective. As a result, IoT data is being lost in all the chaos—unable to pass up important data [13,20,21,22]. Our proposed solution to minimize use of power by implementing effective spectrum management method at FREDEVPLA level is a good solution in solving all above mentioned problems for congested IoT environment.

In order to have an estimation of the device lifetime for a given battery capacity, network configuration, and transmission frequency, we need to understand the power consumption of an IoT device during (1) initial turn-on, (2) connected state, (3) re-transmission and (4) sleep/listen state. To the best of our knowledge, literature research is more focused on optimizing energy efficiency through different scheduling sachems, which is not on solving the cause [4,5,10,11,12,23,24]. Instead, this work is based on empirical evaluation of real deployments, which shed light on the relation between power consumption and spectrum management in congested IoT environment.

The physical test-bed of our testing sets our paper apart from other resource management simulations. We pieced together IoT weaknesses and developed a way to choose a channel with lower interference which resulted in lower power usage within the IoT sensor level. It is important to mention that this work mainly focus on minimizing power utilization of remote IoT sensors not there coordinators (CN). So, power consumed for energy scan purpose is not under consideration in our result.

## 3. System Model

In any conventional IoT communication environment there are three major actors: IoT sensor or sender, coordinator (CN), and destination. Spectrum and power management problem is common at a IoT sensor level. Unlike other works, in our proposal we try to solve these two issues by solving problems created due to congestion. As we understood from other existing studies the main reason for high power consumption is due to difficulty to access channel which is due to bad spectrum management.

We proposed use of FREDEVPLA board in between the coordinator and destination. Its functionalities are: (1), carefully follow and manage spectrum related parameters of overall network. (2), Decide which coordinator can do energy scan and decide when to do scanning. (3), Decide when to change communication channel of the coordinator. The memory capacity of FREDEVPLA allows us to save all the connected CN and its child’s nodes information in a form of table called child table.

As seen in Figure 3, the first step that the FREDEVPLA does is checking the type of data received from any of its coordinators. If the data is DisAssociation Request (DisAss_Req) the FREDEVPLA will delete the mentioned device from its child table list. However, if the received data is Association Request (Ass_Req) it will update its child table list accordingly. If none of the above, the data is forwarded to its destination or saved for future use.

We assume an IoT sensor is distributed randomly to collect specific data such as temperature, light, diabetic sugar levels, A1C etc. The IoT sensor collects the data and transmits it through the CN to its destination. A FREDEVPLA will receive the data packets from various sources of CNs and FREDEVPLA software will then determine whether the data packet should forwarded to the destination or processed by itself. Most of the data packets sent from IoT CN to FREDEVPLA are processed by FREDEVPLA itself. If the data packet is targeted for another IoT device then the IoT CN will send it directly to its destination.

### Energy Scan Algorithm

Algorithm 1 shows how we manage the energy scan between FREDEVPLA and CN. If CN spend >5 s without any activity then the FREDEVPLA will send energy scan instruction to that specific CN. Within the instruction the energy scan time is also mentioned, which various depending on the traffic history. When received scan request from FREDEVPLA the CN will broadcast a notification for all its IoT devices and start energy scan. Once the CN receives the energy scan result it will select a channel with lowest energy and forward it to the FREDEVPLA. After selected channel has been transmitted, the CN returns to conventional communication processing. Here we like to remind our readers that all the commanding and programming is taking place from the FREDEVPLA side. This will allow CN and IoT device to concentrate on communication only.

Below we have the general instructions inside FREDEVPLA if we consider a conventional network, once the channel is selected during initial start-up it will remain constant. This adjustment allowed data to be processed rapidly and faster transmission. It is mentioned in reference [11] that, among main factors of power consumption in IoT communication, retransmission and queue delay are the highest, which is caused because of busy channel. So, solving the busy channel will improve the use of power and overall network performance. It is seen in our result that the busy channel power consumption is significant in the congested IoT network. We solved this problem by avoiding use of busy channel in the IoT environment of the connected devices. As a result we minimized the power consumption significantly and increased the network performance.
**Algorithm 1:** Energy Scan Communication between FREDEVPLA and CN
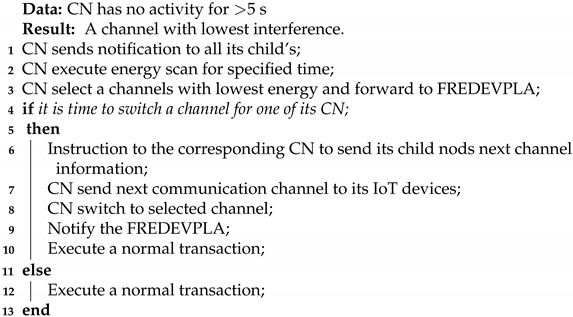


Deciding when to do energy scan, who can do it and when to use the scanned results are among the major novelty of this algorithm. It is known that these three questions have significant impact on almost any channel management algorithm for IoT network. So, the way how we solve these questions will determine the final output of our resources management algorithm. In our algorithm we focused on answering these three questions by executing at an IoT CN level even if the final result is for the IoT devices. The reason why we decided to execute from CN level is to avoid major IoT device limitations such as memory, power, processing speed, etc.

To avoid frequent switching of the channel we implemented minimum throughput level as a requirement which is link quality value, RSSI ≥ −85 dbm. Hence, it is suggested from experimental measurements that a link with RSSI equal or above −85 dBm could be used as a threshold for reliable link estimate [22,25].

As mentioned previously, the focus of this work was to minimize the network power consumption by solving spectrum management, which is one of the main problems facing an IoT network of today’s world. What we did is minimize power consumption from the perspective of busy channel problem solving which is the main reason for power consumption in congested IoT environment.

## 4. Test Set Up

The base setup of our test bed considers the behavior of congested IoT communications. We focus on three communication states of IoT device power consumption: (1) connected state, (2) sleep/listing state, and (3) re-transmission state. Beside these three main focuses we also analyze the power consumption at synchronization and paging state of a device. Our power measurement device is used for both power supply to the IoT device and as a current and voltage measurement. It provides up to 5 V with a high resolution current measurement with a sampling rate up to 4000 samples per second in the range from 1 μA to 5 A.

Furthermore, the measurements were performed soon after the IoT device was turned on, so interference from neighboring IoT device is high. Thus, the main component of the denominator is noise, so we simulated the noise to any regular congested IoT environment. We use a laptop to track both the communication and the power measurement device. Besides measuring power consumption, we also track communication time, retransmission, packet loss and throughput. Overall, we have sent about 2000 packets, 50% of these experiments were run using the conventional network and 50% using the FREDEVPLA, our scheme.

The experiment consists of 20 IoT sensor nodes uniformly deployed in an area of 10–60 M and connected with three collector nodes (MC1321 x EVKRM20). Every sensor can communicate with a full data rate of 50 kbps to the collectors in all available licensed channels. Collectors are placed at the center of the experiment area with only one destination FREDEVPLA. Packet arrival rate is random for every sensor in the network. To perform power consumption measurements and estimate the battery life of the IoT nodes, our power analyzer is next to FREDEVPLA, the voltage level is set constant and the current consumption is measured. FREDEVPLA is placed in the laboratory together with the power analyzer. The power analyzer supplies voltage to all IoT devices and senses the current drawn during the test phase at a sampling rate of 100 Hz. Some of important network setup futures are listed below in Table 1.

The tests have been carried out using CodeWarrior embedded software development studio in a C++ programming environment. Application implemented are built on NXP (Freescale) standard which has IEEE 802.15.4 as a base standard.

Figure 4 represents our experimentation setup. The experiment integrates 3 collectors (CN) connected to the channel coordinator (FREDEVPLA), which is shown to the left of the computer in Figure 4. At the bottom of the same figure, the freescale launchpads that simulate IoT devices are shown. Furthermore, the setup includes a power analyzer that measures the power consumption of each device and provides power to IoT devices. After all the IoT devices were powered, they connected wirelessly with the collectors and send/receive data. Collectors are connected directly to the FREDEVPLA via usb port. FREDEVPLA hooked-up to the computer to see the data transaction for analysis purpose.

First we created a conventional network using three constant channels for three different coordinators. This allowed us to see the power consumption of the conventional network and the result helped us to compare our results. It is clearly seen in our result comparison part that the conventional way of communication channel usage is doing noting to reduce power consumption in a congestion IoT network.

## 5. Test Results and Analysis

After implementing our method in real environment, these were the 4 significant results of our experiment: (1) interference and channel management for all IoT devices from FREDEVPLA is simple and easy (2) successfully measured how much power consumed in connected and sleep statues of IoT device (3) during the lifetime of an IoT device most of its power is consumed in sleep mode and (4) power consumption of IoT device minimized significantly. We executed 20 successive tests using 20 IoT sensors. During each experiment, each IoT sensor would send around 100 data packets. All the information displayed below is the randomly selected tests to send 100 packets per sensor.

### 5.1. Communication States

The two major states considered in our work are: (1) Connected state where the IoT device access the channel to transmit data packet to its coordinator we called it TX state. Includes synchronization, transmission/reception(TX/RX), listening and release. Synchronization (SYNC) is performed by the IoT device to re-synchronize with the network whenever it exits from the Idle state. TX/RX is a period where IoT device receive and transmit packet data. For a short period of time the IoT device listen after every TX/RX and if not received any notification it will enter to idle state this whole process is called listening and release.

### 5.2. Idle State or Sleep State

The idle or sleep state is when the IoT device is only listening. Figure 5 shows both the connected and idle states. Where T_idle represents release phase of connected state.

During the testing of our experiment, no matter how many packets congested the network, the TX/RX process consumes same amount of power.

Figure 6 shows the experimental current traces for a connected state of IoT device for 100 healthcare data packets with load of 20 byte each. The high current consumption is also recorded during these intervals in which the device is actively transmitting, receiving or listing the channel for possible messages from the CN.

It is clear that the power consumption of an IoT device in connected state depends on the time interval spent during each TX/RX. Which is highly affected by the spectrum management mechanism. In other words, if we minimize the TX/RX delay by effective spectrum management using FREDEVPLA the overall performance of our network is better and less energy consuming. However, as the TX/RX interval of IoT device increases, its power consumption also grows to some degree due to the increased network congestion.

Every data transmission is followed by a reception interval where the IoT device waits for possible acknowledgement packets from its CN. In the current traces, the data Tx are preceded and followed by peaks of current consumption, as illustrated in Figure 7. Such peaks are recorded during control signaling traffic. The actual data transmission causes a lower peak and lasts longer. So, adding more controls at the IoT device level to manage spectrum means increase in the average power consumption. That is why we do not recommend adding any channel control configurations at an IoT device level. More than its simplicity, managing configuration for IoT from FREDEVPLA minimize challenges of memory usage, possess speed, and resource management.

The power consumption with our proposed method is always efficient than the normal/standard method minimum by 30% as shown in result comparison part. This is due to our proposed spectrum management scheme always performed separately at the FREDEVPLA level in condition where there is high traffic load.

The plot in Figure 8 show experimental current traces with periodic paging/listing for idle state.

Here it is important to mention that the majority of IoT device’s lifetime is spent on idle state and note that this mode do not have a specific time duration, thus we present the power consumption rather than the energy. Even if the idea behind idle state is to save power, most of the IoT devices do paging for short time intervals during this time to check any waiting messages. The device exits the idle state when the configured time expires or when new CN data are waiting. Regardless of the configured time interval, paging process consumes constant amount of power every single step, seen in Figure 9.

### 5.3. Power Consumption during Initial Turn-On

Interestingly, when the IoT device turns on for the first time, we recorded slight extra energy consumption compared to a device exits from sleep mode. This may be due to active scan, communication parameters setup, or initial control message exchange between the IoT device and its CN.

Figure 10 shows the energy consumption during initial single IoT device turn on. Note that these results recorded for very short micro second, so we present the power consumption rather than the time by zooming the interval. This is a simple proof that how much it is important to deactivate some unnecessary default configuration in our IoT environment before adding the device to the network.

### 5.4. Retransmission Power Consumption

Our test result proved, after implementing spectrum management algorithm at FREDEVPLA to improve the power use for congested IoT environment, the retransmission power consumption decreased significantly. Since our method efficiently managed communication channels, success-fullness of low power consumption is high. In other words, the average power consumption of the IoT device is very less compared to the conventional way of IoT environment. Figure 11 present the power spent for retransmission in a conventional network. This is power spent for retransmission attempt. From the figure, it is clear that the conventional method spends significant amount of power for retransmission purpose in a congested IoT environment. Obviously, if the environment get more congested the number of retransmission will be increased which results in even more power consumption.

In our implementation this power consumption is almost not visible, very low. This comparison is essential for how the proposed method avoids retransmissions and power consumption of IoT devices simply by managing channels at FREDEVPLA level. Furthermore, we like to remind our readers that the result presented in Figure 11 is power spent only for trying retransmission in a network with out FREDEVPLA.

This opens the opportunity to vastly improve the IoT network system by opening up avenues to begin testing an array of managing power and transmission from spectrum angle.

## 6. Result Comparison

The same four major communication states (inactive time, Tx, SYNC, and ReTransmission) analyzed in our result portion are used to compare the power consumption of our method with the base standard. Minimizing the power consumption for these four states of communication affects the overall power consumption significantly.

As shown in Figure 12, the power consumption in standard (Traditional) mechanism is four times higher for retransmission, three times higher for synchronization processes and 30% more consumption for packet transmission compared to our proposed solution. Additionally, our mechanism spend >10% power in the idle/sleep state compared to the standard communication for the same test scenario. Which is a good improvement of our solution on the achieved network performance, since our mechanism executed the same processes in less time and remains in a sleep state for longer time. As a result it saves up to half of the total power consumed in the same network environment and increases the lifetime of IoT device by almost 2× compared to the same IoT device in a standard network.

Furthermore, it is noted that the power consumption for our method and the standard network increases linearly over the interval of connected time which also increases. However, our method has a much lower consumption than the standard network. It is observed that our method performs better than the standard communication in the congested IoT network. It is important to clarify that when we say the standard/traditional in this document we referring to any IoT application which uses IEEE802.15.4 standard as a base for its communication.

We also compared power consumption of successfully received packets without retransmission of our work with the results collected in conventional congested IoT and it is clear that our method is more power saving and has a high network performance.

## 7. Conclusions

To minimize power waste at IoT sensor during network congestion a spectrum based power management system is proposed. The system scans channel conditions before packet routing and utilizes the least congested channel for IoT packet routing.

We used a Freescale Freedom Development Board (FREDEVPLA) as a channel coordinator in our solution. While coordinating channel allocation at the FREDEVPLA level, our system configures MAC and Physical layer of IoT devices in a way that it can have the outstanding power utilization based on the operating network in active, idle and sleep modes compared to the basic IoT standard. A prototype of our solution has been set up by using a number of freescale launchpads that act as IoT devices and can simulate a congested network. A designated FREDEVPLA device was programmed and utilized to execute the proposed spectrum based power management in a congested IoT network scenario. Energy consumption by a channel during connected state, idle state, initial start-up and retransmission over congested IoT networks are measured for a test duration of 6 h. Test data show that battery life of IoT devices using proposed spectrum based power management increases by at least 30% more than non-spectrum based power management methods embedded within IoT devices itself. Additionally, the proposed system eliminates the need for packet retransmission, decreases latency, saves lot of memory for IoT coordinator, improves overall IoT network performance and, more importantly, reduces the risk of losing communication packets. A detailed analysis in this paper will help further research in channel scanning and spectrum based power control.

## Figures and Tables

**Figure 1 sensors-21-02681-f001:**
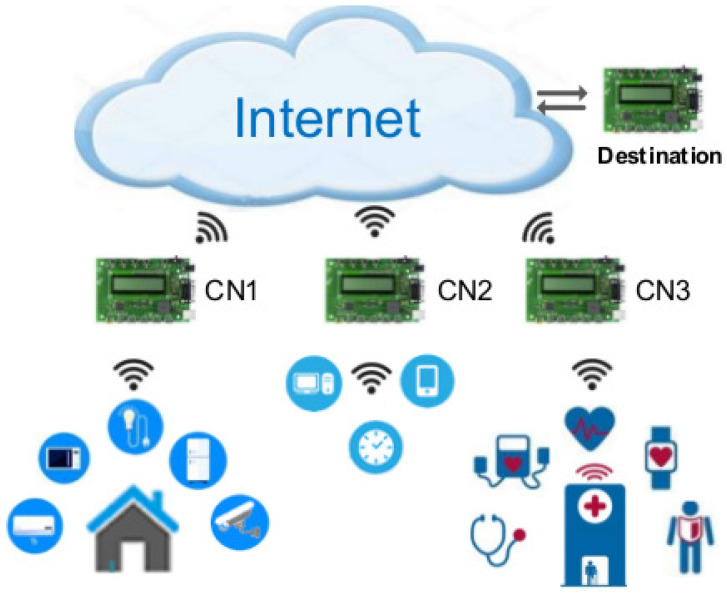
Conventional IoT environment.

**Figure 2 sensors-21-02681-f002:**
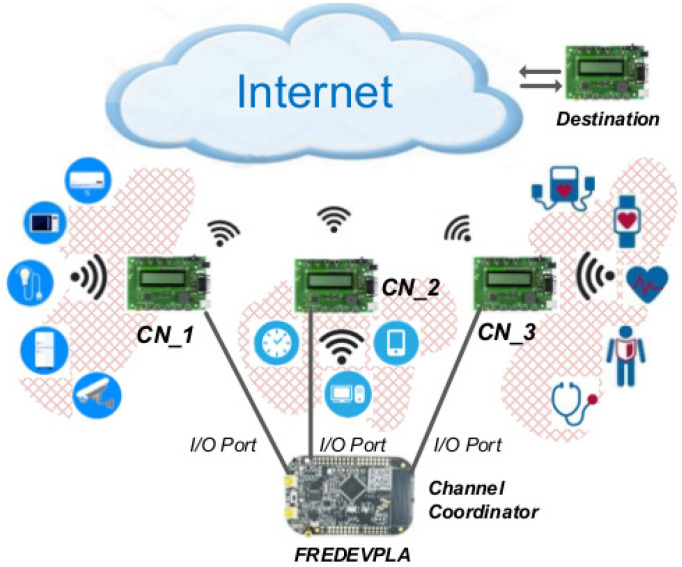
Proposed solution.

**Figure 3 sensors-21-02681-f003:**
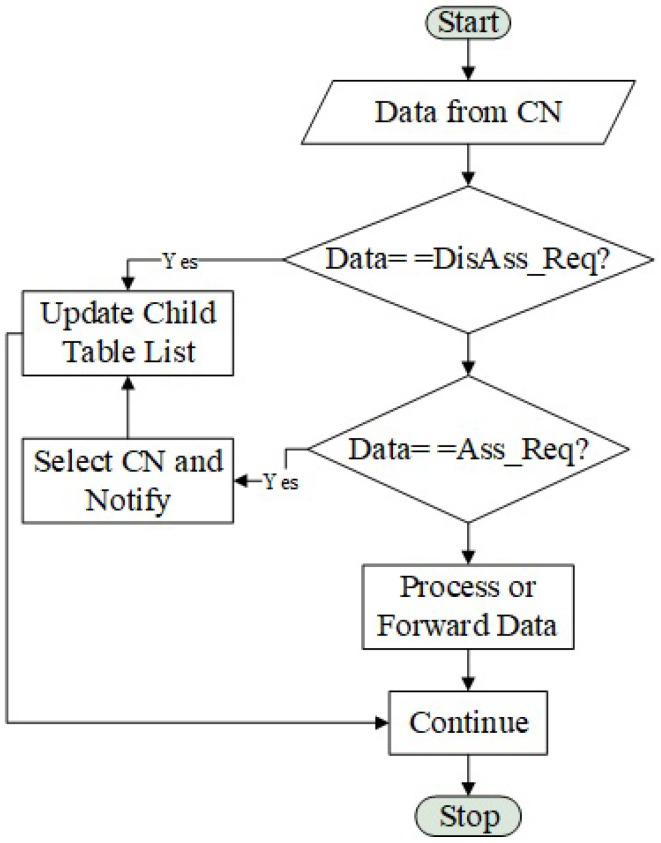
Instruction followed by FREDEVPLA.

**Figure 4 sensors-21-02681-f004:**
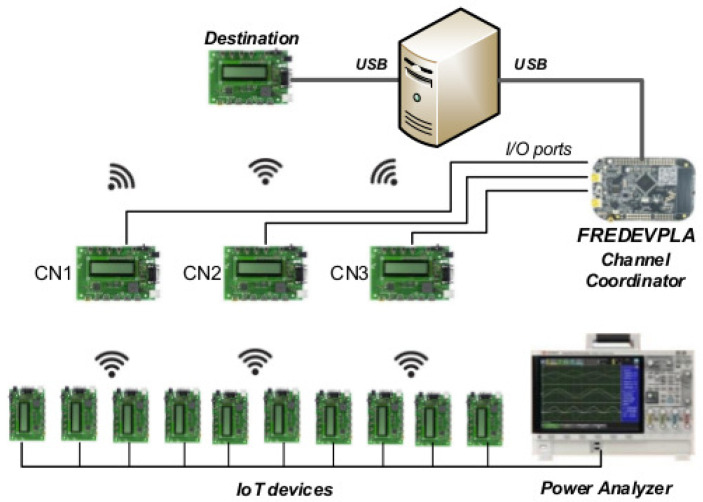
Physical test setup.

**Figure 5 sensors-21-02681-f005:**
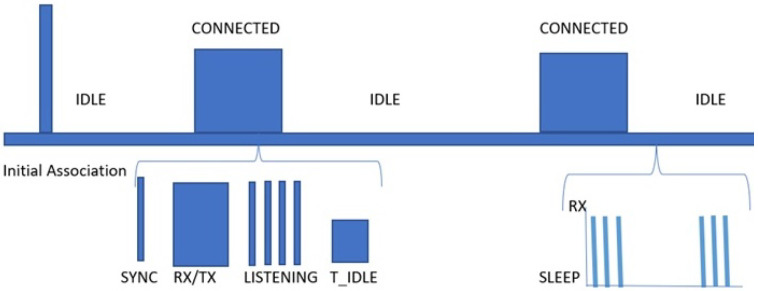
Display of results.

**Figure 6 sensors-21-02681-f006:**
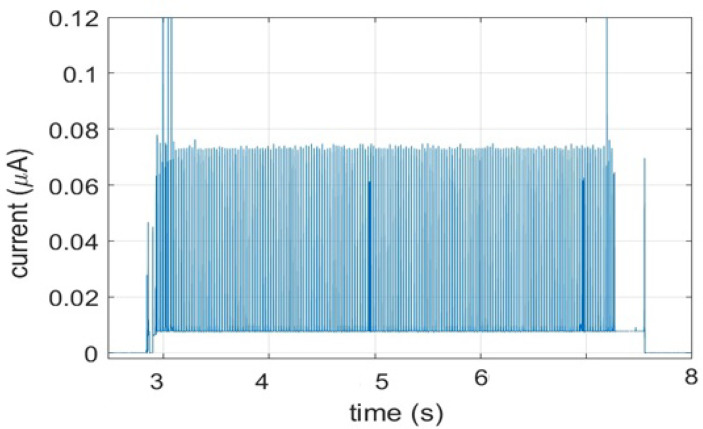
Connected state power consumption.

**Figure 7 sensors-21-02681-f007:**
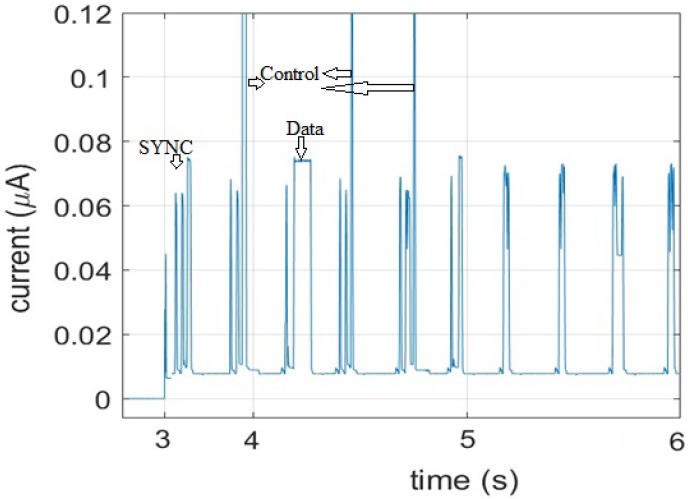
Data and control power consumption.

**Figure 8 sensors-21-02681-f008:**
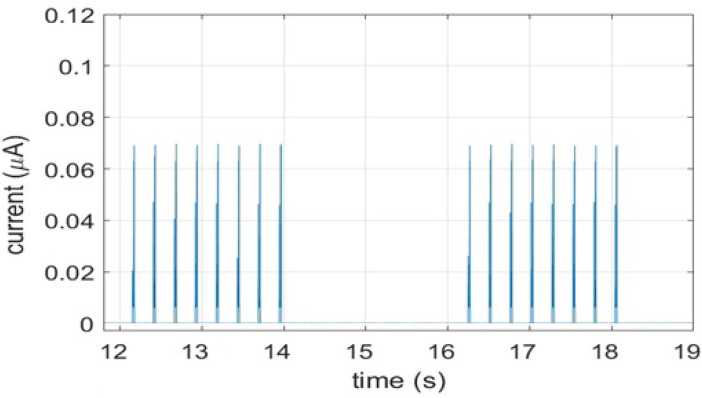
Sleep status consumption.

**Figure 9 sensors-21-02681-f009:**
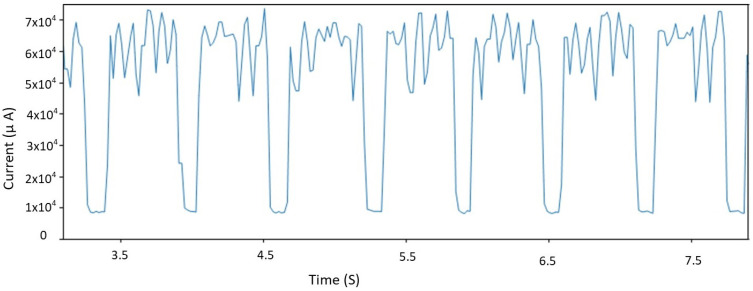
Paging status consumption.

**Figure 10 sensors-21-02681-f010:**
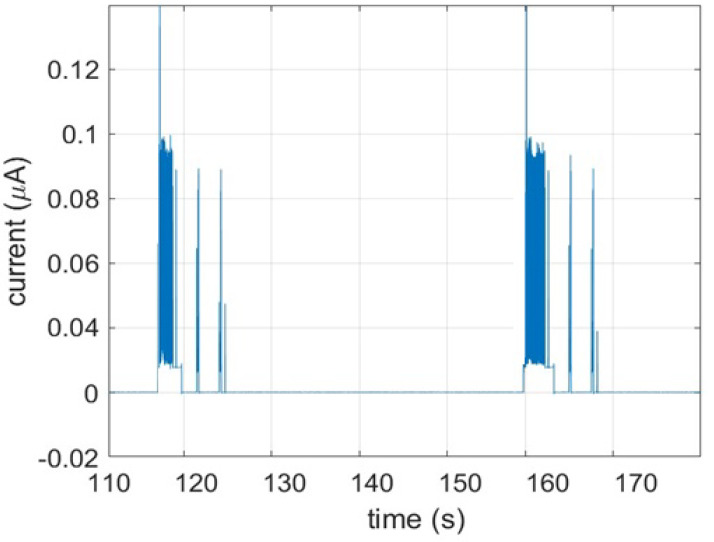
Initial startup power consumption.

**Figure 11 sensors-21-02681-f011:**
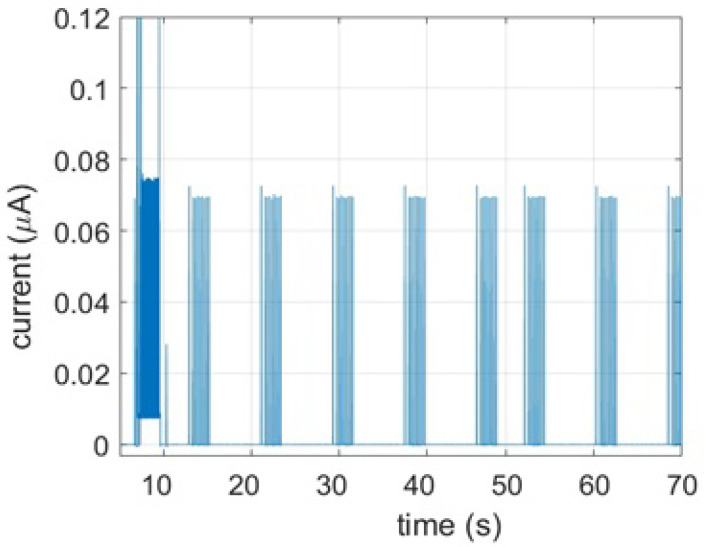
Re-transmission power consumption in conventional congested IoT environment.

**Figure 12 sensors-21-02681-f012:**
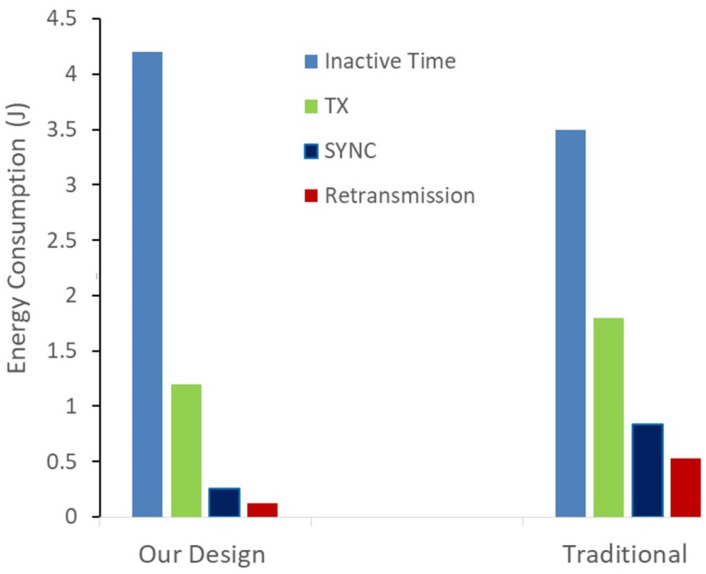
Comparison power consumption.

**Table 1 sensors-21-02681-t001:** Major network setup parameters.

Number of channels used	16
Data rate of the channel	250 kbps
Execution duration	6 h
Frequency	2.4 GHZ

## Data Availability

Not applicable.

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
