# Peer review of "Spectrum Based Power Management for Congested IoT Networks"

_sensors, 2021, doi:10.3390/s21082681_

Round 1

Reviewer 1 Report

In this paper, the authors proposed a spectrum based power management solution that scans channel conditions when needed and utilizes the lowest congested channel for IoT packet routing. According to the experiments conducted, the proposed solution shows certain potentials in reducing power consumption while maintaining quality of communication at more hardware level. It is a meaningful piece of fieldwork, but revisions are suggested to further improve its soundness. 

  1. The congestion control mechanism is widely studied in a variety of researches from different angles. The authors are suggested to review and compare the relevant works in this area and highlight the advantages (and possible limitations) of the spectrum-based solution proposed.
  2. Frequent switching of the channel could be problematic as well. The authors are suggested to explain the impact and propose effective measures for error monitoring. 

Author Response

On behalf of my coauthors, I would like to thank you for the opportunity to revise and resubmit our manuscript sensors-1128343, entitled “SPECTRUM BASED POWER MANAGEMENT FORCONGESTED IoT NETWORKS.” We found the reviewers’ comments to be helpful in revising the manuscript and have carefully considered and responded to each suggestion. In the majority of cases, we were successful in incorporating the reviewers’ feedback into our revised manuscript. 

Reviewer 2 Report

The authors propose a spectrum based power management solution for congested IoT networks. Channel scanning and selection are used for effectively managing power utilization by minimizing retransmission power.

The paper is well-structured and well-written. The results are clearly presented.

However, to further improve the paper some changes/integrations are needed. Comments are reported in the following:

- lines 72-86: these details should not be included in the  introduction section. An overview of the proposed system  is enough. The introduction section may be improved by highlighting the innovation aspects of the proposed solution with respect to the state of the art.

-Figure 2 is not clear. The cannel to be scanned is the one between the sensors and the coordinator. the cannel shall be better highlighted in the figure. It is not clear how the “channel coordinator” is connected to the sink and how they interact.

- line 119: what about the energy consumption for channel information gathering?  Do the authors estimate it? this shall be taken into account for an effective estimation of the real decrease of power consumption within the network. It seems that only the decrease of power consumption of the sensors nodes has been analyzed. If so, the sentence shall be rephrased.

- Figure 4: the destination is the Sink? Please clarify this.

- The analysis has been carried out from the IoT sensors’ perspective. This is fine if the objective is to decrease the sensors' power consumption. What about the CN power consumption (traditional vs proposed model)? Moreover, if we focus on the overall power consumption, also the channel coordinator shall be taken into account. Also it would be interesting to evaluate the overall energy consumption of the system

- The increase of the QoS level has not been properly discussed. It has only been cited as a consequence of the power consumption reduction (delay reduction). The authors should better rephrase the sentences regarding the QoS.

I recommend this paper for publication, after a revision addressing the aforementioned  comments.

Author Response

(The authors gave the same response as above.)

Reviewer 3 Report

The submitted manuscript sensors-1128343 proposed a spectrum based power management solution with channel scanning for IoT packet routing. Several comments are provided as follows. Comment 1 The authors state that the developed system provides the best power utilization based on the operating network (e.g., at Line 15 and Line 70). The authors may need to provide experimental results or formulations to effectively validate that the system can achieve the best performance. Some effort on this issue should be further addressed. Comment 2 It seems that some devices in Figure 4 is not the real deployed devices (e.g., the Power Analyzer, the board FREDEVPLA). The authors may have to explain why these devices are not shown in the Physical test Setup with their real in-situ configuration but using some edited pictures. Comment 3 The Algorithm 1 is the main proposed channel scanning algorithm for energy management. The novelty or contribution of this algorithm is not clear. Since the power management algorithm is proposed as the main objective of this paper, the authors may have to clearly present the contributions of the proposed algorithm. Comment 4 In Section 6. Result Comparison, the traditional mechanism is compared. The author indicated the compared mechanism as standard (traditional). What is the exactly standard mechanism is compared? Is it an existing mechanism that has been used or can be referenced with a citation? More information about the compared baseline can be provided. Comment 5 There are so many grammar issues and typos existing in the current manuscript (for example, the sentence at Line 134-Line 136). The authors need to carefully revise them to improve the readability of this paper.

Author Response

(The authors gave the same response as above.)

Round 2

Reviewer 2 Report

The authors have updated the manuscript according to the reviewers' comments, improving the overall quality of the paper and clarifying the requested points.

I recommend this paper for publication in the current form.

Author Response

On behalf of my coauthors, I would like to thank you for the opportunity to revise and resubmit our manuscript sensors-1128343, entitled “SPECTRUM BASED POWER MANAGEMENT FORCONGESTED IoT NETWORKS.” We found the reviewers’ comments to be helpful in revising the manuscript and have carefully considered and responded to each suggestion. 

Thank you again for your consideration of our revised manuscript.

Reviewer 3 Report

Please find the detailed review report in the attached PDF file.

Author Response

On behalf of my coauthors, I would like to thank you for the opportunity to revise and resubmit our manuscript sensors-1128343, entitled “SPECTRUM BASED POWER MANAGEMENT FORCONGESTED IoT NETWORKS.” We found your comments to be helpful in revising the manuscript and have carefully considered and responded to each suggestion. In the majority of cases, the answers for your question are found in the document we highlighted for you in the revised manuscript. There was one suggestion related to adding more experimental results( Q# 1 & 4) that we were unable to implement because it would have necessitated changing the nature of the study to address the your concern and it is out of our objective.  Our initial plane was to do more result comparation and presenting more experimental results but because of the Covid-19 pandemic our labs are closed. Which enforced us to limit our experiment objective to the basic IoT standard which we did before the pandemic.  

Regarding your Comment 2  which read as "The contributions of the proposed paper is still not clear in this paper. In other words, what are the novelties of the
proposed method? For a paper with 16 pages,....."  We  highlighted in red but in case if that don't seem to you the contribution of the paper we respect you comment but that's all what we are presenting as our contribution in this paper.

Thank you again for your consideration of our revised manuscript.

Round 3

Reviewer 3 Report

In this revised manuscript, the authors address the concerns in the previous review report.

Author Response

Thank you again for your consideration of our revised manuscript.

Sincerely,

Kedir Mamo, PhD.

University of Texas at Dallas & UABC

800 W. Campbell Road, Richardson, TX, 75080-3021

United States

+1 469-987-1726

[email protected]
